# Impact of childhood traumatic brain injury on fitness for service class, length of service period, and cognitive performance during military service in Finland from 1998 to 2018: A retrospective register-based nationwide cohort study

Julius Möttönen[1]*, Ville T. Ponkilainen[2], Ville M. Mattila[1,2], Ilari Kuitunen[3,4]

1 Faculty of Medicine and Life Sciences, Tampere University, Tampere, Finland, 2 Department of Orthopedics and Traumatology, Tampere University Hospital, Tampere, Finland, 3 Institute of Clinical Medicine, University of Eastern Finland, Kuopio, Finland, 4 Department of Pediatrics, Kuopio University Hospital, Kuopio, Finland

* Julius.mottonen@tuni.fi

**Data Availability Statement:** According to the latest update in the Finnish legislation (552/2019)

## Abstract

## Introduction

Traumatic brain injury (TBI) can cause neuronal damage and cerebrovascular dysfunction, leading to acute brain dysfunction and considerable physical and mental impairment long after initial injury. Our goal was to assess the impact of pediatric TBI (pTBI) on military service, completed by 65–70% of men in Finland.

## Methods

We conducted a retrospective register-based nationwide cohort study. All patients aged 0 to 17 years at the time of TBI, between 1998 and 2018, were included. Operatively and conservatively treated patients with pTBI were analyzed separately. The reference group was comprised of individuals with upper and lower extremity fractures. Information on length of service time, service completion, fitness for service class, and cognitive performance in a basic cognitive test (b-test) was gathered from the Finnish Military Records for both groups. Linear and logistic regression with 95% CI were used in comparisons.

## Results

Our study group comprised 12 281 patients with pTBI and 20 338 reference group patients who participated in conscription. A total of 8 507 (66.5%) men in the pTBI group and 14 953 (71.2%) men in the reference group completed military service during the follow-up period. Men in the reference group were more likely to complete military service (OR 1.26, CI 1.18–1.34). A total of 31 (23.3%) men with operatively treated pTBI completed the military service. Men with conservatively treated pTBI had a much higher service rate (OR 7.20, CI 4.73–

on the secondary use of patient information in Finland, any register data can not be made available for researchers living outside Finland. The law states that the any data can not be made publicly available without compromising patient privacy. Researchers registered in Finland or in Finnish educational institutes, can access the data by applying permit from Findata the Finnish data authority (Care Register For Health Care - data) and from the Data Protection Director of the Finnish Defence Forces (Finnish Military Records - data). Contact information for applying the data: The Finnish Health and Social Data Permit Authority Findata. info@findata.fi.

**Funding:** IK has received research funding from Maire Taponen Foundation (https://www.mairetaposensaatio.fi) and from Yrjö Jahnsson Foundation (https://www.yjs.fi). VP and IK have received funding from the Päivikki and Sakari Sohlberg Foundation (https://pss-saatio.fi). Funders did not play any role in the study design, data collection, analysis, decicion to publish or preparation of the manuscript.

**Competing interests:** The authors have declared that no competing interests exist.

11.1). In the pTBI group, men (OR 1.26, CI 1.18–1.34) and women (OR 2.05, CI 1.27–3.36) were more likely to interrupt military service than the reference group. The PTBI group scored 0.15 points (CI 0.10–0.20) less than the reference group in cognitive b-test.

## Conclusions

PTBI groups had slightly shorter military service periods and higher interruption rate than our reference-group. There were only minor differences between groups in cognitive b-test.

## Introduction

Traumatic brain injury (TBI) is defined as a disruption of brain function or structural damage to the brain caused by external force. Based on clinical findings and imaging results, TBI is categorized as mild, intermediate, and severe [1–3]. Pediatric traumatic brain injury (pTBI) is a major cause of mortality, morbidity, and disability with up to 280 per 100 000 children worldwide being diagnosed with TBI annually [4]. Boys are notably over represented in the incidence of TBI, especially after the age of 4 [4]. In Finland, the incidence of mild pTBI increased continuously from 1998 through 2018, whereas the incidence of severe TBIs that require acute neurosurgical interventions remained the same [5,6].

TBI causes neuronal damage and cerebrovascular dysfunction that can lead to acute brain dysfunction [7]. The cerebrovascular dysfunction is believed to continue for months or even years after the initial injury and can accelerate brain aging and increase cognitive dysfunction [7]. Regardless of the severity, pTBI has been shown to cause considerable physical and mental impairment long after initial injury [8]. A recent 2021 meta-analysis revealed that children with TBI suffer for a long period with cognitive fatigue and loss of cognitive function in tasks that require prolonged brain function [9]. Cognitive impairments, such as concentration, attention, and memory deficits as well as emotional disorders (anxiety, depression), are the cause of considerable harm later on in working life, academic studies, and social relations [8,10]. In addition to cognitive and emotional damage to the brain, it has been estimated that almost 20% of patients who are admitted to hospital with pTBI will have some lifelong physical disabilities, which can include other accident-related injuries [11].

In a recent Swedish register-based study, pTBI increased the risk for hospitalized psychiatric conditions, disability pension, and premature mortality. Moreover, the risk increased with multiple TBIs and the severity of the TBI [12]. In a follow-up survey, pTBI patients reported poor performance at school, lower quality of life, and a more hopeless future than the general population [13].

In Finland, all 18-year-old males are called-up for evaluation to serve in the Finnish military, and females are eligible to participate voluntarily. On average, 65–70% of males and 1–4% of females complete military service. The long-term impact of pTBI to one's fitness has not been studied before. The goal of our study was to assess the impact of previous pTBI on fitness for service classification, length of military service period, and cognitive performance in the Finnish Defence Forces when compared to an orthopedically injured reference population (Fig 1).

## Materials and methods

We conducted a retrospective register-based nationwide cohort study in Finland. The data were assembled using two nationwide registers: the Finnish Care Register for Health Care and Finnish Military Records. The study period was from January 1998 to December 2018 [14,15].

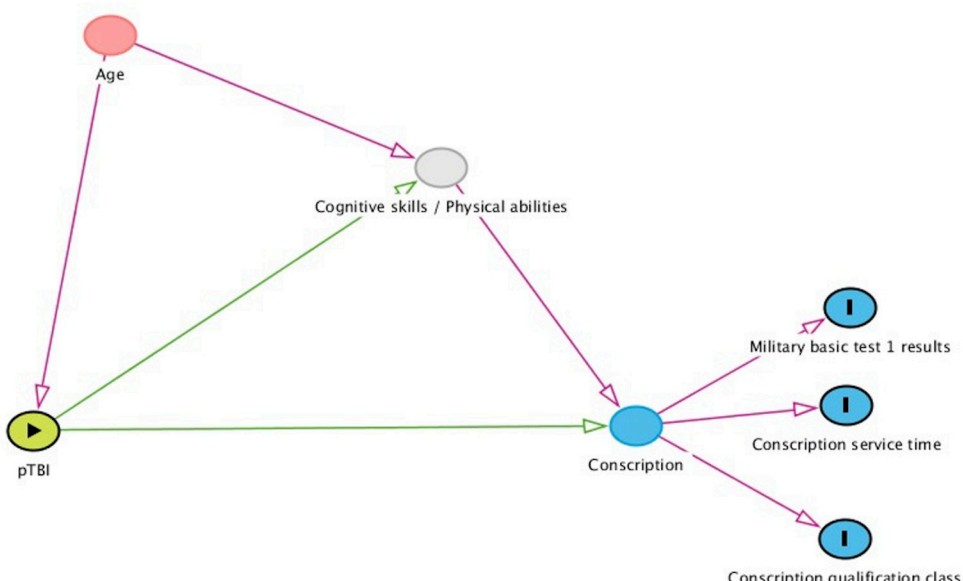

**Fig 1. Hypothetical illustration of the impact of pTBI on fitness for service class, length of service period, and cognitive performance in military service.**

### Finnish care register for health care data

Pediatric patients (aged 0 to 17 years) with TBIs were included from the Finnish Care Register for Health Care. The Register includes information on all specialized health care (secondary and tertiary level units) visits, procedures, and hospitalizations in Finland. The register has been shown to have excellent coverage and validity [16]. The patient group (later referred to as the pTBI group) included all emergency department visits and hospitalizations with ICD-10 diagnostic code S06*. Operatively treated pTBIs were analyzed separately, and the group was formed using certain Nordic Medico-Statistical Committee Classification of Surgical Procedures codes (Finnish version). A reference group was formed from children with upper and lower extremity fractures (later referred to as the reference group). The reference group was formed to mimic similar risk-taking behavior as those patients with pTBI. The complete list of diagnosis codes and operation codes used in all groups are presented in S1 Appendix.

### Finnish military records data

Information on length of military service, cognitive performance, and fitness for service class was gathered from Finnish Military Records. As conscription is compulsory for men in Finland, the register only contains information on those women who participate in conscription voluntarily. Conscription is usually performed at the age of 19 to 20 years, but in special cases entry between the ages of 18 and 29 years is also possible. In the present study, we included all 18-year-olds who had completed their conscription, had time to complete their service and had been given a fitness for service class during the follow-up period. Individuals older than 25 years at the end of the follow-up were handled as being released from peacetime military/civilian service. We chose the age of 25 due to the limited study period and because most conscription postponements are education or health related, which are usually dealt with in 3 to 6 years. People younger than 25 years who did not have a fitness for service class were excluded due to uncertainty over the completion of their military service. In Finland, military service is divided into periods of 6, 9, and 12 months, depending on the training involved. The fitness

for service class that is given at call-up is divided into five categories A, B, E, C, and T based on the conscripts' physical and mental health (A = suitable without restrictions, B = suitable with restrictions on mental or physical performance, E = postponement due to medical reasons, C = relief for peacetime for medical reasons, T = found too mentally fragile for military service). Conscripts who are categorized as A or B also have the right to apply for civilian service instead of military service. Currently, approximately 7% of conscripts decide to complete civilian service every year [17]. Between the years 1998 and 2018, the number of men who began conscription varied between 30 685 and 24 095, and the completion percentage was 83.8% on average. We were, however, unable to verify whether an individual was released from military service or ended up in civilian service. Therefore, men with fitness for service class C and those who completed civilian service are handled together.

The service class classification during the conscription is done by licensed physician with guidance from conscription health examination manual [18]. The exemptions are based on ICD-10 diagnose codes and the manual presents the possible service class options for each diagnose code separately. None of the pTBI group or reference group diagnosis codes directly justifies exemption from military. To be released from military service a person must have the specific diagnose code/codes and the injuries/diseases indicated by the code/codes must significantly reduce mental or physical capacity. Only the diagnose code S06.0 (concussion), cannot solely be the reason for exemption from the service.

Cognitive performance is evaluated in a cognitive basic test (b-test) at 6 to 8 weeks after commencing military service. The b-test comprises three subtests of verbal, arithmetic, and visuospatial reasoning. Each subtest comprises around 40 multiple-choice questions, ranging from easy to complicate. Scores from the subtests are then converted to standard nines and range from 0 to 9 with mean of 5.00 and standard deviation of 1.96. The final score given to a conscript is the average of the three subtests. The results of the b-test are then used to evaluate the intellectual skills of the conscripts and to determine their final posting. B-test scores from 0 to 3 usually indicate rejection from more demanding 9 to 12 month duties.

## Statistical analyses

We combined information from the Finnish Care Register for Health Care and the data from Finnish Military Records to form our complete study population using the patients' unique identification number (Fig 2). All comparisons in the analyses were done between the pTBI group and the reference group as well as for the pTBI group and the operatively treated pTBI group. All analyses were done separately for men and women due to the voluntary nature of women´s military service. We used linear regression to analyze the difference between the groups of b-test results with 95% confidence intervals (CI). Welch two sample t-test was used to evaluate a statistically meaningful difference between the groups in b-test. We used age adjusted logistic regression to calculate odds ratio with 95% CI on military service (started or not), fitness for service class (A vs others), length of military service, and for military service interruption rate. The length of military service was calculated comparing 6 months to longer periods as a whole and separately to 9 months and 12 months. All statistical analyses were done using R version 4.0.5 (R Foundation for Statistical Computing, Vienna, Austria) [19].

## Ethics statement

We received permission to access the Care Register for Health Care from Findata (the Finnish data authority) (permission number: THL/2058/14.02.00/2020). Permission to use Finnish Military Records was acquired from the Data Protection Director of the Finnish Defence Forces. According to Finnish research legislation, ethical committee evaluation is not required

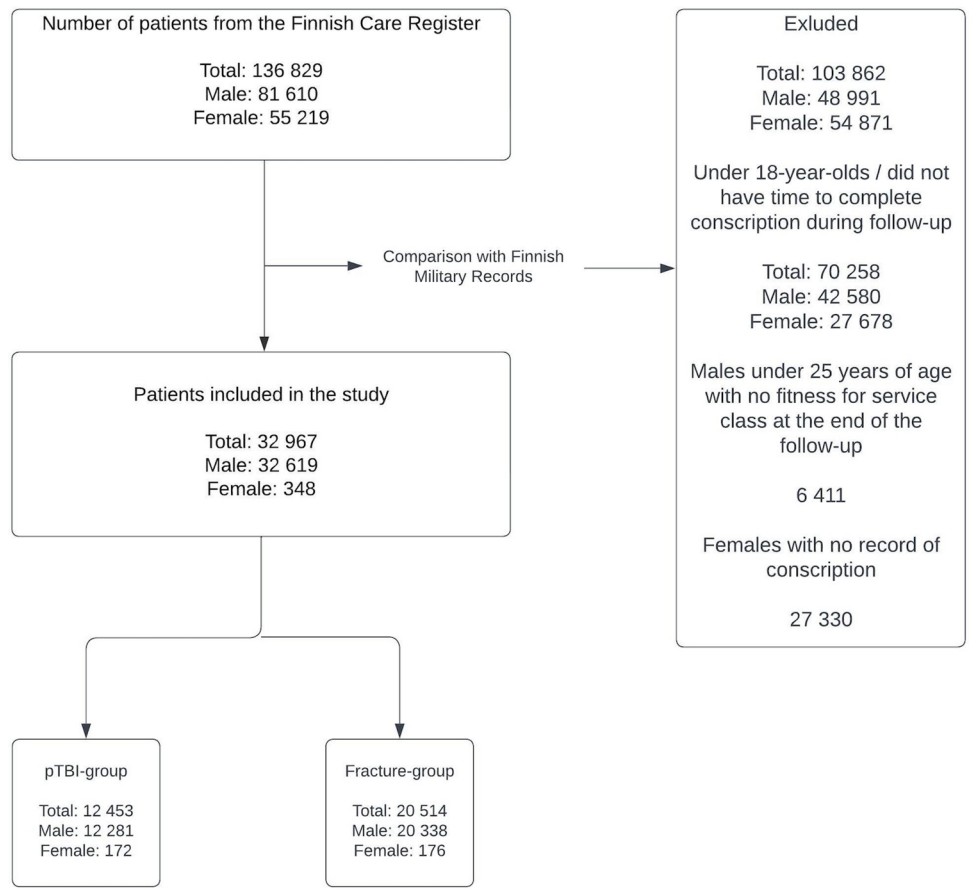

**Fig 2. Flowchart of the formation of the study population.**

for register-based retrospective studies involving pseudonymized human participants' data. Therefore, according to the Finnish research legislation, consent was neither required from the participants or parents/guardians. We accessed the Care Register for Health Care data in 15th of February 2020 and Finnish military records in 15th of May 2023. Statistics Finland and Finnish Defence Forces had pseudonymized the data before our access and the authors did not have access to the pseudonymization key. The authors could not identify any individuals during or after the data collection. The data was handled and stored in a safe remote-controlled environment, which required two-phase identification at every login.

## Results

A total of 12 281 men who sustained pTBI and 20 338 men who fractured an ankle or wrist during childhood participated in military service during the 21-year follow-up period. In total, 88 men with pTBIs were operatively treated. Mean age at the time of injury was 11.8 years for men in the pTBI group and 12.6 years in the reference group. A total of 172 women with pTBI and 176 women with childhood orthopaedical injury voluntarily participated in military service during the follow-up period. The mean age at the time of injury in women in the pTBI group was 11.6 years and 12.2 years in the reference group. No women with an operated pTBI participated in the conscription (Table 1).

In total, 10 422 (84.9%) men in the pTBI group and 17 631 (86.7%) men in the reference group began military service during the study period. Furthermore, men in the pTBI group

**Table 1. Number of people who participated in military service, their service interruption/completion rate, mean age at the time of the injury, and cognitive b-test results between the pTBI group, the reference group, operative treatment, men, and women.** Calculated odds ratios and p-values between pTBI group and reference group as well as conservatively and operatively treated pTBI.

|  | pTBI group | Reference group | Operatively treated pTBI | PTBI group / reference group | Conservatively/ operatively treated pTBI) |
|---|---|---|---|---|---|
| MEN |  |  |  |  |  |
| Participated call-up | 12 281 | 20 338 | 88 | - |  |
| Started service | 10 422 (84.9%) | 17 631 (86.7%) | 39 (44.3%) | OR 0.84, CI 0.78–0.89 | OR 6.55, CI 4.29–10.1 |
| Interrupted service | 1 915 (15.6%) | 2 678 (13.2%) | 8 (9.09%) | OR 1.23, CI 1.16–1.32 | OR 0.85, CI 0.41–1.07 |
| Completed service | 8 507 (69.2%) | 14 953 (73.5%) | 31 (35.2%) | OR 0.81, CI 0.76–0.86 | OR 1.18, CI 0.94–2.44 |
| Mean injury age | 11.8 | 12.6 | 13.7 | - | - |
| Performed b-test | 8 248 (67.1%) | 14 180 (69.7%) | 29 (33.0%) |  |  |
| Mean b-test result | 4.74 | 4.89 | 5.07 | P ≤ 0.001 | P = 0.265 |
| WOMEN |  |  |  |  |  |
| Participated call-up | 172 | 176 | - | - | - |
| Started service | 172 | 176 | - | - | - |
| Interrupted service | 58 (33.7%) | 35 (19.9%) | - | OR 2.07, CI 1.28–3.40 | - |
| Completed service | 114 (66.3%) | 141 (80.1%) | - | OR 0.48, CI 0.29–0.78 | - |
| Mean injury age | 11.6 | 12.2 | - | - | - |
| Performed b-test | 140 (81.4%) | 135 (76.7%) | - | - | - |
| Mean b-test result | 5.17 | 5.29 | - | P = 0.571 | - |

Note: Mean injury age is calculated in years. OR = Odds Ratio. CI = Confidence Interval.

were less likely to participate in military service than men in the reference group (OR 0.84, CI 0.78–0.89), and men with conservatively treated pTBI were more likely to start military service than men with operatively treated pTBI (OR 6.55, CI 4.29–10.1). A total of 8 507 (69.2%) men in the pTBI group and 14 953 (73.5%) men in the reference group completed their military service. However, 1 915 (15.6%) men in the pTBI group and 2 678 (13.2%) men in the reference group interrupted their military service. Men in the pTBI group were more likely to interrupt military service than men in the reference group (OR 1.23, CI 1.16–1.32). We found no evidence of a difference between the conservatively treated and operated pTBI in the military service interruption rate (OR 0.85, CI 0.41–1.07) (Table 1).

All the women who voluntarily participated in conscription started their military service. Moreover, 114 (66.3%) women in the pTBI-group and 141 (80.1%) in the reference group completed their military service. A total of 58 (33.7%) women in the pTBI group and 35 (19.9%) women in the reference group interrupted their military service. Moreover, women in the pTBI group were more likely to interrupt their military service than women in the reference group (OR 2.07, CI 1.28–3.40) (Table 1).

The most common length of military service time for men in both the pTBI group and the reference group was 6 months. Men in the pTBI group were slightly more likely to have a shorter 6-month military service period than men in the reference group (OR, 1.08, CI 1.02–1.14). Men with operatively treated pTBI were more likely to serve a shorter 6-month period than men with conservatively treated pTBI (OR 0.39, CI 0.17–0.81). For both groups of women, the most common service period was 12 months. There was no observable difference in the length of service periods between the two groups of women (OR 0.91, CI 0.33–2.43) (Tables 2 and 3).

Total of 8 248 (67.1%) men in the pTBI group and 14 180 (69.7%) men in the reference group performed the cognitive b-test. Of those who completed military service, 1 973 (11.2%) men in the reference group and 1 076 (10.3%) men in the pTBI group had missing results. In

**Table 2. Service periods of those who completed military service between the pTBI group, the reference group, and operative treatment as well as between genders.** Calculated odds ratios between pTBI group and reference group as well as conservatively and operatively treated pTBI.

| Military service period | PTBI group | Reference group | Operatively treated pTBI |
|---|---|---|---|
| MEN | | | |
| 6 months | 3 783 (44.5%) | 6 399 (42.8%) | 21 (67.7%) |
| 9 months | 1 170 (13.8%) | 2 285 (15.3%) | 3 (9.68%) |
| 12 months | 3 551 (41.7%) | 6 269 (41.9%) | 7 (22.6%) |
| WOMEN | | | |
| 6 months | 8 (7.02%) | 10 (7.09%) | - |
| 9 months | 22 (19.3%) | 32 (22.7%) | - |
| 12 months | 84 (73.7%) | 99 (70.2%) | - |

**Table 3. Calculated odds ratios between pTBI group and reference group as well as conservatively and operatively treated pTBI.**

| Military service periods | Odds ratios between PTBI group and reference group | Odds ratios between Conservatively / operatively treated |
|---|---|---|
| MEN | | |
| 6 months vs. 9 and 12 months | OR 1.08, CI 1.02–1.14 | OR 0.39, CI 0.17–0.81 |
| 6 months vs. 9 months | OR 1.15, CI 1.06–1.25 | OR 0.46, CI 0.11–1.35 |
| 6 months vs. 12 months | OR 1.05, CI 1.00–1.12 | OR 0.36, CI 0.14–0.81 |
| WOMEN | | |
| 6 months vs. 9 and 12 months | OR 0.91, CI 0.33–2.43 | - |
| 6 months vs. 9 months | OR 1.12 CI 0.37–3.31 | - |
| 6 months vs. 12 months | OR 0.86 CI 0.30–2.31 | - |

the b-test, men with childhood pTBI scored 0.15 points (CI 0.10–0.20, $P \leq 0.001$) less than the reference group. However, there was no remarkable difference between conservatively treated pTBI and operated pTBI in b-test 1 results (0.33 points, CI -0.32 to 0.99, P = 0.265). Two men in the operated pTBI group completed service and had no recorded results. Total of 140 (81.4%) women in the pTBI group and 135 (76.7%) women in the reference group performed the b-test and 18 (10.2%) women in the reference group and 14 (8.14%) in the pTBI group completed military service and had missing results. In women, the b-test results did not differ considerably between the two groups (0.12 points CI -0.28 to 0.52, P = 0.571) (Table 1).

The most common fitness for service class was A in both groups and between men and women. For men with operated pTBI, the most common fitness for service class was C/Civilian service. Men in the pTBI group were less likely to have fitness for service class A than men in the reference group (OR 0.78, CI 0.75–0.83). Men with conservatively treated pTBI were more likely to have fitness for service class A than men with operated pTBI (OR 5.22, CI 3.29–8.59). For women, there were no notably difference in those women who had fitness for service class A in the pTBI group and the reference group (OR 0.55, CI 0.27–1.09) (Table 4).

## Discussion

In this study, we found that men with previous pTBI were able to participate and successfully complete military service. However, when compared to our reference group, men in the pTBI group were more likely to be exempted from military service or participate in civilian service.

**Table 4. Number of people in different fitness for service classes between pTBI group, reference group, and operative treatment as well as between genders.**

| Qualification class | pTBI group | | Reference group | | Operatively treated pTBI | |
|---|---|---|---|---|---|---|
| | Men | Women | Men | Women | Men | Women |
| A | 8 089 (65.9%) | 149 (86.6%) | 14 368 (70.6%) | 162 (92.0%) | 23 (26.1%) | - |
| B | 735 (5.98%) | 6 (3.49%) | 1 119 (5.50%) | 4 (2.27%) | 7 (7.95%) | - |
| C | 3 456 (28.1%) | 17 (9.88%) | 4 849 (23.8%) | 10 (5.68%) | 58 (65.9%) | - |
| T | 1 (0.00%) | 0 | 2 (0.00%) | 0 | 0 | - |
| Total: | 12 281 | 172 | 20 338 | 176 | 88 | - |

The difference was greatest with operatively treated pTBI. Both men and women with pTBI were more likely to interrupt their military service than men and women in the reference groups, and people with pTBI had slightly shorter military service periods.

Before starting military service, an individual is required to be in good physical and mental condition. In the conscription health examination manual, there are numerous post-injury disorders and emotional disorders that are acceptable reasons for conscripts to receive an exemption from conscription or military service [18]. Even though our reference group consisted mainly of direct injuries to the limbs, our results indicate that men with pTBI have slightly more of these exemption disorders that lead to a lower military service rate. In both men and women, the military service interruption rate was higher in the pTBI group. Furthermore, the proportion of men in the fitness for service class A category was notably higher in the reference group than in the pTBI group. The literature supports our findings as it has been reported that pTBI causes long-term mental and physical impairment that can affect the fitness of an individual [8,10,12]. Although the pTBI group had a lower military service rate, it still falls between the national average of 65–70% of Finnish men completing their military service.

The length of military service in the Finnish military is determined by the type of training the conscript undergoes. Typically, those persons who have a military service period of 6 months partake in basic crew duties that require less mental and physical strain, whereas persons with longer service periods of 9 or 12 months undergo more demanding officer, non-commissioned officer, or specialist training [20]. One of the factors that determines the length of service time is the b-test, which tests the cognitive abilities of a conscript. Both men and women in the pTBI group scored slightly lower in the b-test than the reference group, and the pTBI group had slightly more 6-month periods of military service than the reference group. This suggests that even though TBI can drastically affect mental and physical health, individuals are still able to serve in the military almost as well as people without a background of head injury.

When men with conservatively treated pTBI were compared with men who underwent acute neurosurgical interventions after pTBI, however, the differences were greater. Indeed, the military service rate was markedly lower in the operatively treated pTBI group. The service interruption rate was also higher in the operatively treated pTBI group, but the finding is unreliable due to the large confidence interval. In addition, the fitness for service category C was clearly more common in the operatively treated pTBI group. All these findings refer to better physical and mental abilities in the conservatively treated group. This finding is supported by the prior literature that has reported that moderate and severe pTBIs can cause long-term and life-long physical and mental impairment [21]. Still, the fact that there are people serving in the military who have had operatively treated pTBI is interesting. One of the justifications for being released from peacetime military service can be a specific brain injury with operative treatment. One might think that TBI that needed neurosurgical intervention as a child/adolescence would have such severe long-term impact to the mental or physical capacity of an

individual that it would be highly unlikely to be able to perform military service. The really unexpected finding was that the operatively treated pTBI group had higher mean b-test results than the conservatively treated pTBI group. This might be partly explained by the fact that a large part of this group had been eliminated during conscription or early in their military service before the test was performed. Therefore, those people who were still in military service or who had completed it were the ones with fewer post-injury disabilities.

Our study brings important new information on the long-term impact of childhood/adolescence TBI. As this topic has been only scarcely researched before, our study suggests that pTBI has negative impact to fitness for years. Also, even though severe TBIs usually are reason for exemption from military service, there were quite a few operatively treated pTBI-patients that started the service and completed it. The conscription health examination manual does not categorize TBIs as mild, intermediate, or severe [18]. The classification is only done between the ICD-10 codes S06.0-S06.8 [18]. It is therefore in the hands of the physician to evaluate the severity and late effects of brain injury to functional ability and service suitability. Based on our study, it could be beneficial to add a more detailed instructions to the manual about assessing suitability for the service with pTBI background, especially those with moderate to severe brain injuries.

Access to the nationwide Finnish Care Register for Health Care was a major strength of our study. The register includes all visits and operations through all health care levels, including primary, secondary, and tertiary levels. Furthermore, the register contains information on ICD-10 codes and operation codes and has been shown to have excellent coverage [22,23]. This database supports the generalizability of our findings to the Finnish pediatric population. In Finland, conscription is compulsory for all male citizens aged 18 years. Therefore, it is highly unlikely that any men aged 18 years at the time of the follow-up are missing from the database. A weakness in our study is the lack of information on the clear Glasgow Coma Scale rated severity of the TBI. In addition, our results are not thoroughly generalizable in the female population because of the voluntary nature of the participation of women in military service. We also had some missing data regarding the cognitive b-test results that might have affected the resuts.

## Conclusion

People with pTBI were able to complete their military service almost as well as the orthopedically injured reference population. Although the PTBI group had slightly shorter service periods and higher interruption rates, there were only slight differences in cognitive b-test scores between groups. Men with operatively treated pTBI had the highest percentage of short service periods and most early releases from peacetime military service.

## Supporting information

**S1 Appendix. ICD-10 Diagnose codes and operation codes used in the study.**
(DOCX)

## Acknowledgments

We would like to complement Peter Heath for the language review of this article.

## Author Contributions

**Conceptualization:** Ville T. Ponkilainen, Ville M. Mattila, Ilari Kuitunen.

**Data curation:** Julius Möttönen.

**Formal analysis:** Julius Möttönen.

**Supervision:** Ville T. Ponkilainen, Ville M. Mattila, Ilari Kuitunen.

**Writing – original draft:** Julius Möttönen.

**Writing – review & editing:** Ville T. Ponkilainen, Ville M. Mattila, Ilari Kuitunen.

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
