## [Decision Letter · Decision Letter 0]

17 Mar 2024

PONE-D-23-42327Impact of childhood traumatic brain injury on fitness for service class, length of service period, and cognitive performance during military service in Finland from 1998 to 2018: A retrospective register-based nationwide cohort study.PLOS ONE

Dear Dr. Möttönen,

Thank you for submitting your manuscript to PLOS ONE. After careful consideration, we feel that it has merit but does not fully meet PLOS ONE’s publication criteria as it currently stands. Therefore, we invite you to submit a revised version of the manuscript that addresses the points raised during the review process.

We look forward to receiving your revised manuscript.

Kind regards,

Sreeram V. Ramagopalan

Academic Editor

PLOS ONE

Journal Requirements:

Reviewers' comments:

Reviewer's Responses to Questions

**Comments to the Author**

1. Is the manuscript technically sound, and do the data support the conclusions?

Reviewer #1: Partly

2. Has the statistical analysis been performed appropriately and rigorously? 

Reviewer #1: No

3. Have the authors made all data underlying the findings in their manuscript fully available?

Reviewer #1: Yes

4. Is the manuscript presented in an intelligible fashion and written in standard English?

Reviewer #1: Yes

5. Review Comments to the Author

Reviewer #1: Dear Editor,

The article by Mottonen et al report results from a data linkage study of healthcare records with military service records for individuals who as children suffered a traumatic brain injury (pTBI) compared to children who suffered arm or leg fractures.

I was uncertain why this research is needed. I would also suggest more clarity on the criteria, timelines and order in which tests are given, and individuals are assessed for service. I got the impression that individuals with pTBI shouldn’t be serving at all according to guidelines but nonetheless some do. This can potentially introduce bias into the sample. A more robust discussion on the potential biases is needed.

Detailed comments:

1) It is not clear why this research is needed. How will it impact future fit for service guidelines or medical recommendations for paediatric patients with these injuries?

2) Please be clear in the methods - if conscription guidelines explicitly exclude people with pTBI.

3) The discussion states: “there are numerous post-injury disorders and emotional disorders that are acceptable reasons for conscripts to receive an exemption from conscription or military service” later it goes on to say “ major operations are part of the exclusion criteria in the military handbook” Does this mean that recruits lie or are unaware of their past TBI history / major operation as a child? Does this bias the sample who make it into the service and who will eventually finish service? Is the research trying to elucidate this?

4) Please include a table with the OR – I would suggest adding them to Table 1 and Table 2 respectively.

5) I note there was no matching between groups, but the number of females was surprisingly similar in both groups. Can the authors comment on incidence rates of TBI being similar to those of fractures? It is possible that randomly the groups are the same size, but it just really jumped out at me.

6) Age is identified in Figure 1 as the main (only) confounder of the association, but it is not adjusted for in the logistic regression. Please provide rationale.

7) What is the sample size for the cognitive impairment test – if it is given 6-8 weeks after enrolment and the authors suggest some people might drop out by then ; the sample size might be smaller for this analysis.

8) I think it would be valuable to present b-test score by length of service –larger differences between groups might be observed.

9) What difference in the b-test score would be clinically meaningful? Or in the military what cut off are used to determine final posting.

Abstract:

1) Reference category includes a range of lower limb fractures wider than just ankle and wrist.

6. PLOS authors have the option to publish the peer review history of their article (what does this mean?). If published, this will include your full peer review and any attached files.

Reviewer #1: No

---

## [Author Response · Author response to Decision Letter 0]

22 Apr 2024

DETAILED COMMENTS:

1)

It is not clear why this research is needed. How will it impact future fit for service guidelines or medical recommendations for paediatric patients with these injuries?

Author response: Thank you for this comment. We agree that this information was missing. We have now assessed this thoroughly in the manuscript. Please see, Introduction: Paragraph 4. and Discussion: Paragraph 5.

2)

Please be clear in the methods - if conscription guidelines explicitly exclude people with pTBI.

Author response: The reviewer 1 is correct. The Methods lacked detailed information how the service class is determined. We have now added paragraph to explain the reasoning behind the classification Simply, any form of a TBI is not an automatic exclusion from the service, everyone and every case is evaluated individually. Please, see Methods: Paragraph 4 and Discussion: Paragraph 4.

3)

The discussion states: “there are numerous post-injury disorders and emotional disorders that are acceptable reasons for conscripts to receive an exemption from conscription or military service” later it goes on to say, “major operations are part of the exclusion criteria in the military handbook” Does this mean that recruits lie or are unaware of their past TBI history / major operation as a child? Does this bias the sample who make it into the service and who will eventually finish service? Is the research trying to elucidate this?

Author response: Thank you for this comment. Please, see above. We have explained the exclusion criteria in a more detailed fashion. 

4)

Please include a table with the OR – I would suggest adding them to Table 1 and Table 2 respectively.

Author response: We have now added all the ORs to tables as suggested. For the tables to be as clear as possible, we have created a new table for ORs regarding service length (Table 3). Previous Table 3 has now been moved to be Table 4 Please, see Results: Table 1, 3 and 4. 

5)

I note there was no matching between groups, but the number of females was surprisingly similar in both groups. Can the authors comment on incidence rates of TBI being similar to those of fractures? It is possible that randomly the groups are the same size, but it just really jumped out at me.

Author response: Thank you for pointing this out. We were surprised as well. It is completely coincidental that the groups were almost exactly same size. 

6)

Age is identified in Figure 1 as the main (only) confounder of the association, but it is not adjusted for in the logistic regression. Please provide rationale.

Author response: Thank you for this comment. It was indented to run all analyses with age adjustment. This statement has now been added to the Methods section. We also checked the logistic regression analyses and simply by honest mistake there were few analyses done without age adjustment. New analyses were performed, and results are now updated entirely regarding the ORs. There were only minor differences, and this error did not affect the final interpretation of the results. Please, see Methods: Paragraph 6, Results: Tables 1 and 3.

7)

What is the sample size for the cognitive impairment test – if it is given 6-8 weeks after enrolment and the authors suggest some people might drop out by then ; the sample size might be smaller for this analysis.

Author response: We agree with the Reviewer. Sample size has now been added to Table 1 and Results. Indeed, there were some individuals who completed military service, but had missing data from the cognitive b-test. We have added this as limitation. Please, see Results: Paragraph 5 and Table 1, Discussion: Paragraph 6.

8)

I think it would be valuable to present b-test score by length of service –larger differences between groups might be observed.

Author response: As you can see from the table below, conscripts with longer service length have better results in this test. Differences between groups are not larger. The test is done before anyone knows how long they will be serving and logically people with better test results will be selected more often to the longer service periods. This is why we chose to report the results in one group. The fact that pTBI group has lower average score is the most important independent factor and the effects are seen in the service length between groups. In addition, we have now calculated P-values for the b-test results using Welch two sample t-test. Please, see Results: Table 1. 

Length of service pTBI group Reference group Difference pTBI group Reference group Difference

 Men Men Women Women 

6 months 4.28 4.40 0.12 CI 0.05-0.20 5.50 5.43 0.07 CI -2.36-2.50

9 months 4.68 4.81 0.12 CI -0.00-0.25 4.74 4.48 0.25 CI -1.19-0.68

12 months 5.34 5.50 0.16 CI 0.09-0.23 5.31 5.60 0.29 CI -0.21-0.80

All 4.47 4.89 0.15 CI 0.10-0.20 5.17 5.29 0.12 CI -0.28-0.52

9)

What difference in the b-test score would be clinically meaningful? Or in the military what cut off are used to determine final posting.

Author response: B-test scores from 0 to 3 usually indicate rejection from more demanding 9 to 12-month duties. Please, see Methods: Paragraph 5.

ABSTRACT:

1)

Reference category includes a range of lower limb fractures wider than just ankle and wrist.

Author response: It is true. There also were shin fractures and fractures of patella. We have corrected this in the manuscript. Please, see Abstract: Paragraph 2 and Methods: Paragraph 2.

---

## [Editor Report · Decision Letter 1]

2 May 2024

Impact of childhood traumatic brain injury on fitness for service class, length of service period, and cognitive performance during military service in Finland from 1998 to 2018: A retrospective register-based nationwide cohort study.

PONE-D-23-42327R1

Dear Dr. Möttönen,

We’re pleased to inform you that your manuscript has been judged scientifically suitable for publication and will be formally accepted for publication once it meets all outstanding technical requirements.

Kind regards,

Sreeram V. Ramagopalan

Academic Editor

PLOS ONE
---

## [Editor Report · Acceptance letter]

7 May 2024

PONE-D-23-42327R1 

PLOS ONE

Dear Dr. Möttönen, 

I'm pleased to inform you that your manuscript has been deemed suitable for publication in PLOS ONE. Congratulations! Your manuscript is now being handed over to our production team.

Kind regards, 

on behalf of

Dr. Sreeram V. Ramagopalan 

Academic Editor

PLOS ONE